# Systemic interleukin 10 levels indicate advanced stages while interleukin 17A levels correlate with reduced survival in esophageal adenocarcinomas

**Karl-Frederick Karstens**[1], **Jan Kempski**[2], **Anastasios D. Giannou**[2], **Erik Freiwald**[3], **Matthias Reeh**[1], **Michael Tachezy**[1], **Jakob R. Izbicki**[1], **Ansgar W. Lohse**[2], **Nicola Gagliani**[1,2,4], **Samuel Huber**[2]*, **Penelope Pelczar**[2]

1 Department of General, Visceral and Thoracic Surgery, University Medical Center Hamburg-Eppendorf, Hamburg, Germany, 2 Section of Molecular Immunology und Gastroenterology, I. Department of Medicine, University Medical Center Hamburg-Eppendorf, Hamburg, Germany, 3 Department of Medical Biometry and Epidemiology, University Medical Center Hamburg-Eppendorf, Hamburg, Germany, 4 Immunology and Allergy Unit, Department of Medicine, Solna, Karolinska Institute and University Hospital, Stockholm, Sweden

☯ These authors contributed equally to this work.

* shuber@uke.de

**Data Availability Statement:** All relevant data are within the paper and its Supporting Information files.

## Abstract

### Introduction

Reflux promotes esophageal adenocarcinomas (EAC) creating a chronic inflammatory environment. EAC show an increasing incidence in the Western World and median survival rates are still low. The main reasons for poor prognosis despite new multimodal therapies are diagnosis of EACs at an already advanced stage and distant metastases. Hence, we wanted to investigate the presence of systemic inflammatory interleukins (IL) and their impact on patient prognosis.

### Material and methods

Systemic expression levels of pro- and anti-inflammatory markers (IL-2, IL-4, IL-6, IL-10, IL-17A and IL-22) in the sera of 43 EAC patients without neoadjuvant radiochemotherapy were measured by flow cytometric analysis. A correlation to clinicopathological data was performed. Log-rank and Cox regression analysis were used to investigate the impact on patient survival. 43 sera of age and gender matched healthy volunteers were used as controls.

### Results

Increased systemic IL-6 (p = 0.044) and lower IL-17A (p = 0.002) levels were found in EAC patients as opposed to controls. A correlation of IL-10 levels with an increased T stage was found (p = 0.020). Also, systemic IL-10 levels were highly elevated in patients with distant metastasis (p<0.001). However, only systemic IL-17A levels had an influence on patient survival in multivariate analysis.

**Funding:** The author(s) received no specific funding for this work.

**Competing interests:** The authors have declared that no competing interests exist.

## Conclusion

Systemic IL-6 levels are increased, while IL-17A levels are reduced in EAC patients compared to healthy controls. In addition, circulating IL-10 might help to identify patients with advanced disease and high IL-17A might indicate a limited prognosis.

## Introduction

Esophageal adenocarcinomas (EAC) have an increasing incidence in the Western World and survival rates are still low with a median 5-year survival of approximately 25% [1]. One of the risk factors for developing EAC is prolonged acid and bile exposure to the distal part of the esophagus causing a state of chronic inflammation [2]. Most of the patients are diagnosed at already advanced stages and response to neoadjuvant chemotherapy remains low [3,4]. Hence, new treatment options like immune therapies are being investigated. However, the role of inflammation on carcinogenesis and prognosis in esophageal cancer is still being debated. In particular, little is known about the influence of systemic cytokines in EACs. Both, pro-and anti-inflammatory cytokines are involved in local and systemic tumor development. Pro-inflammatory cytokines like interleukin- (IL-) 6 or IL-2 mainly attract and activate other inflammatory cells in the tumor microenvironment [5–7]. Both interleukins have been reported to play a role in esophageal squamous cell carcinomas (ESCC) and other gastrointestinal malignancies like gastric cancers [8–15]. However in EACs conflicting results have been reported [16–18]. On the other hand, anti-inflammatory cytokines like IL-4 and IL-10 have a predominantly regulatory function in the immune system [19,20]. Increased as well as decreased levels of IL-4 and IL-10 have been reported in esophageal and gastric cancers and their prognostic value remains uncertain [21–26]. However, one meta-analysis identified serum IL-10 as a negative prognostic marker in several gastrointestinal malignancies including colon and gastric cancers. Though, this study didn't include esophageal cancers [27]. In addition, we recently demonstrated a strong association of IL-10 secreted by regulatory T cells with patients' survival not only in the tumor tissue but also in the unaltered mucosa close to the resection margin representing a local immunological field effect [28]. On the basis of these results, we hypothesized that systemic IL-10 concentration might also correlate with the stage or survival of EACs. Therefore, the aim of this study was to measure systemic levels of IL-10. Other cytokines involved in chronic inflammation are IL-17 and IL-22 [29–31]. Both interleukins have pro- as well as anti-inflammatory properties and their roles in tumor development and progression are still being debated [32–35]. In addition, data regarding the role of systemic IL-17 or IL-22 in esophageal cancers are sparse and their clinical influence in this malignancy is unclear [36].

In summary, only very few studies for systemic cytokines in esophageal cancer especially in EACs have been conducted and conflicting results have been reported. In the light of upcoming potential immune therapies, a deeper understanding of the systemic inflammatory response is necessary. Hence, we investigated serum levels of IL-2, IL-4, IL-6, IL-10, IL-17A and IL-22 as potential biomarkers and examined their correlation with clinicopathological factors and overall survival in EACs. To rule out a potential alteration of the immune response, only patients without neoadjuvant radiochemotherapy were selected.

## Material and methods

### Patients

Study on human sera was approved by the Medical Ethical Committee, Hamburg, Germany (PV3548 and PV4444). Written informed consent was obtained from all patients and healthy

volunteers before study inclusion. All procedures performed in this study involving human participants were in accordance with the ethical standards of the institutional and national research committee and with the 1964 Helsinki declaration and its later amendments or comparable ethical standards. Blood samples were drawn the day before surgery and immediately processed. Only patients with histopathological confirmed EAC and no neo-adjuvant radio or radiochemotherapy were selected. Age, gender, extent of tumor (T stage), metastatic lymph node status (N stage), distant metastases (M stage), tumor infiltration of resection margins (R status), tumor grading (G status), Union internationale contre le cancer (UICC) stage, body mass index (BMI) and survival were analyzed in EAC patients.

Healthy controls (n = 43) were selected from a prospective biobank after matching for gender (40 males, 3 females) and age (median 62.0 years, range 46.0–83.0 years). All volunteers were blood donors and at excellent clinical condition at the moment of blood donation.

## Sample preparation

Whole blood was centrifuged at 2000 rpm for 10 min. After centrifugation the total amount of serum from the 10 mL collection tube was transferred to another tube and stored at −80°C until analysis. For serum cytokine quantification LEGENDplex™ (BioLegend, Koblenz, Germany; Catalog no. 740721) human T helper cell cytokine panel was used. Experimental procedures have been performed according to the manufacturer's protocol. If samples were below the limit of detection (LOD) values were represented by half of the LOD.

## Statistical analysis

Statistical analysis was performed with GraphPad Prism® Software (GraphPad Software, San Diego, CA, USA). The relationship between each cytokine (IL-2, IL-4, IL-6, IL-10, IL-17A and IL-22) and each factor studied was determined using the non-parametric Spearman correlation coefficient. All statistically significant results by Spearman correlation coefficient were furthermore confirmed with the non-parametric Mann–Whitney U-rank sum test. To compare the prognostic effect of interleukin levels the median was used as cutoff and a log rank test was performed for Kaplan-Meier curves. For multivariate survival measurements statistical analysis was performed using SAS for Windows, version 9.4 (SAS Institute Inc., Cary, NC). A Cox regression analysis was done to investigate the effect of variables on overall survival. All cytokines were tested against UICC stage and R status in multivariate analysis. Patients that died within 30 days after surgery were excluded from survival analyses. All statistical tests were considered significant at p<0.05.

## Results

### Systemic cytokines in EAC compared to controls

We first compared serum levels of IL-2, IL-4, IL-6, IL-10, IL-17A and IL-22 in EAC patients with serum interleukin levels in healthy controls (Fig 1). In EAC patients, systemic IL-6 levels were significantly (p = 0.044) elevated with a median of 7.35 pg/ml (range 2.71–454.0 pg/ml) as compared to healthy controls with a median of 5.69 pg/ml (range 2.98–77.94 pg/ml). In addition, systemic IL-17A levels were significantly (p = 0.002) lower in EAC patients demonstrating a median of 47.1 pg/ml (range 3.43–505.7 pg/ml) while controls showed a median of 84.28 pg/ml (range 6.86–320.5 pg/ml). Slightly, albeit not significantly, elevated systemic IL-10 levels (p = 0.265) with a median of 6.45 pg/ml (range 2.05–302.3) and reduced systemic IL-4 levels (p = 0.069) with a median of 3.82 pg/ml (range 0.85–239.3 pg/ml) as opposed to controls, which showed a median of 5.64 pg/ml (range 2.19–45.78 pg/ml) and 5.55 pg/ml (range 1.98–

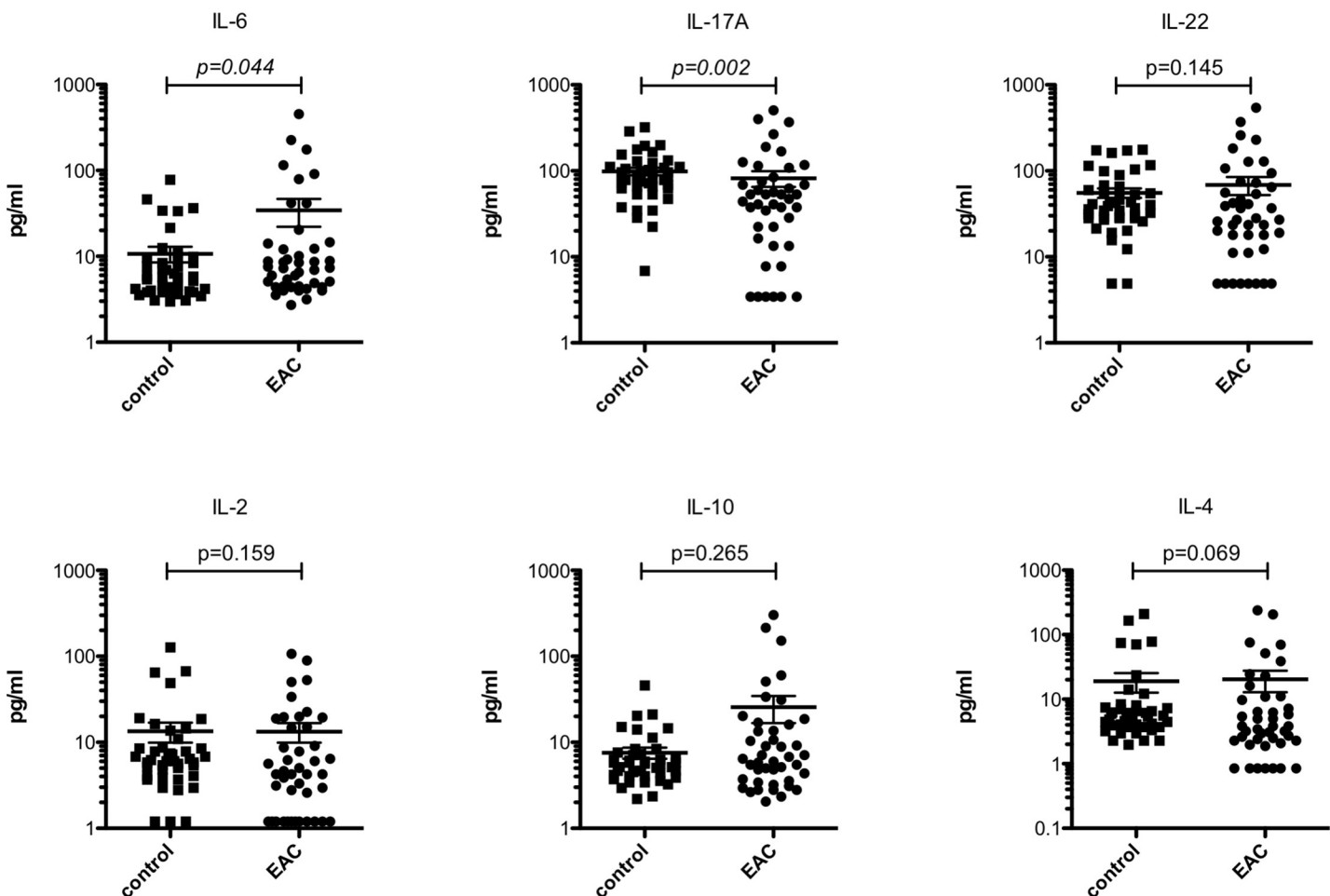

**Fig 1. Systemic IL-6 and IL-17A distinguish EAC patients from healthy controls.** Significant differences (p<0.05) are highlighted in italic. Bars indicate mean with standard error.

210.5 pg/ml), respectively, were found. Moreover, none of the other investigated cytokines demonstrated a significant difference.

## Association of cytokines in EAC with clinical data

First a Spearman correlation was performed to identify potential associations between interleukins and clinical parameters (S1 Table). In this analysis, only IL-10 levels showed a significant positive correlation with T, M and UICC stage as well as mortality.

We then looked into each T, N and M stage in detail. Systemic IL-10 levels significantly (p = 0.020) increased with T stage where T1 stage has a median of 3.2 pg/ml (range 2.05–7.11 pg/ml) while T3 stage has a median of 6.45 pg/ml (range 2.64–214.8 pg/ml). A median difference of 10.36 pg/ml (range 2.79–302.3 pg/ml) between T1 and T4 stage was also observed but failed to reach significance by a small margin (p = 0.081). No difference between T3 and T4 stages was observed. Neither of the other interleukins showed significant differences between T stages (Fig 2). When looking into the N status, none of the investigated interleukins demonstrated a significant alteration between the stages (Fig 3). In patients with distant metastases (M1 stage), systemic levels of IL-10 were significantly (p<0.001) elevated demonstrating a median of 50.96 pg/ml (range 10.68–302.3 pg/ml) as opposed to those without metastases (M0

stage) showing a median of 5.31 pg/ml (range 2.05–214.8 pg/ml). No other significant alterations between M0 and M1 stages for the other investigated interleukins were found (Fig 4).

The different T, N and M stages are comprised in the UICC classification. In the latter classification a stepwise increase of systemic IL-10 levels from stage I with a median of 2.99 pg/ml (range 2.05–7.11 pg/ml) to stage II showing a median of 4.83 pg/ml (range 2.94–18.64 pg/ml) over stage III demonstrating a median of 5.47 pg/ml (range 2.64–214.8 pg/ml) to stage IV with a median of 50.96 pg/ml (range 10.68–302.3) was observed. In comparison to all other UICC stages, the median of stage IV was significantly higher (control to UICC stage IV: p = 0.0001; UICC stage I to IV: p = 0.0012; UICC stage II to IV: p = 0.007 and UICC stage III to IV: p = 0.007, respectively). In addition, a trend towards a significant increase of IL-10 between stages I and III was observed (p = 0.089). However, no additional differences between the other UICC stages were found. None of the other investigated interleukins demonstrated significant alterations between UICC stages (S1 Fig).

## Survival analysis

None of the patients was lost during follow-up. At the end of the study, 18 patients (41.9%) were alive and 25 patients (58.1%) had died. Four patients (9.3%) died within 30 days after

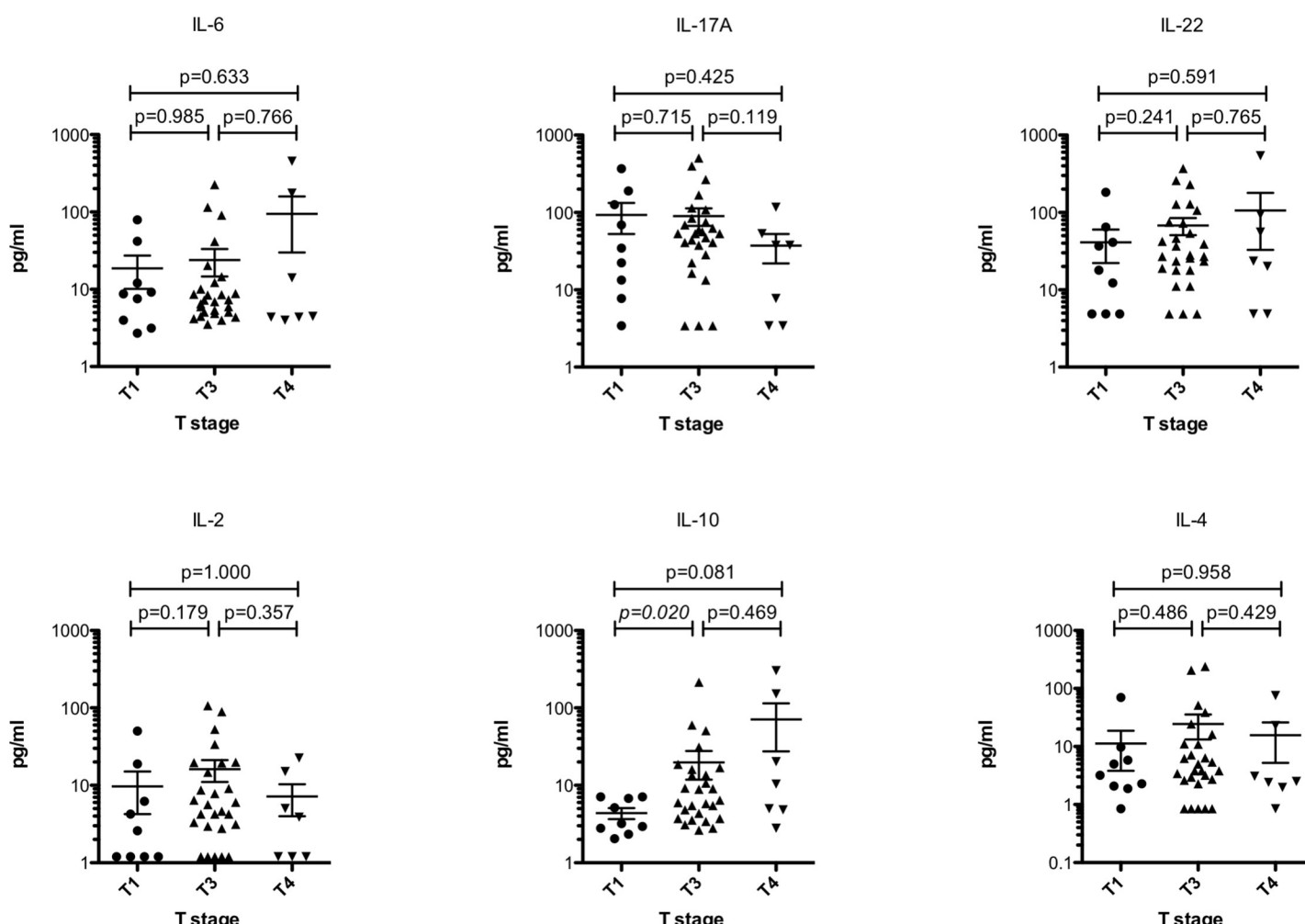

**Fig 2. Systemic IL-10 increases with the progression of T stage.** Significant differences (p<0.05) are highlighted in italic. Bars indicate mean with standard error.

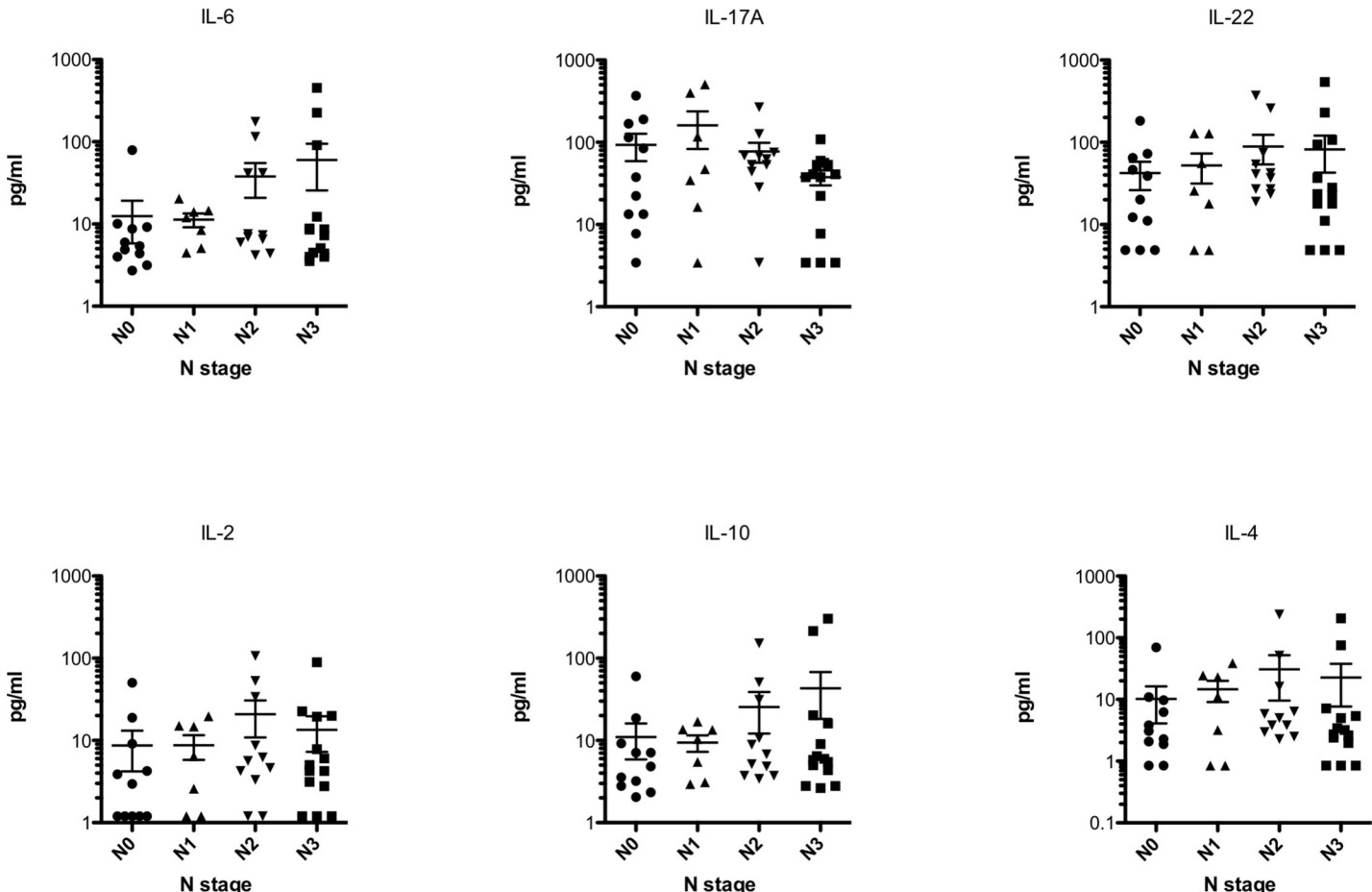

**Fig 3. None of the investigated systemic interleukins significantly changes with N stages.** Since no significant changes were observed bars between groups are not drawn for better visualization.

surgery and were excluded from further survival analysis. Median survival was 18.9 month (range 2.3–62.4 month). None of the patients died for other causes than esophageal cancer. When analyzing the clinical and histopathological data, the occurrence of distant metastases (M1 stage; p = 0.005), residual tumor cells in the resection margin (R1 status; p = 0.005) and UICC stages (p = 0.007) were significantly associated with overall survival. The local extent of the tumor (T stage) failed to significantly affect survival by a small margin (p = 0.051). For further details see Table 1. Surprisingly, neither did IL-10 nor any of the other investigated systemic interleukins demonstrate a significant influence on overall survival in univariate analysis (Fig 5A). However, patients with systemic levels of IL-10 above the median split yielded a median survival of 20.43 month as opposed to patients with IL-10 levels below the median split demonstrating a median survival of 43.13 months.

For multivariate analysis interleukins were tested against UICC stage and R status since both parameters had a strong impact on survival in the latter analysis. However, only IL-17A showed a significant influence on overall survival in multivariate analysis (HR 0.152 95% CI 0.034–0.670) indicating a better survival for patients with low systemic IL-17A levels (Fig 5B).

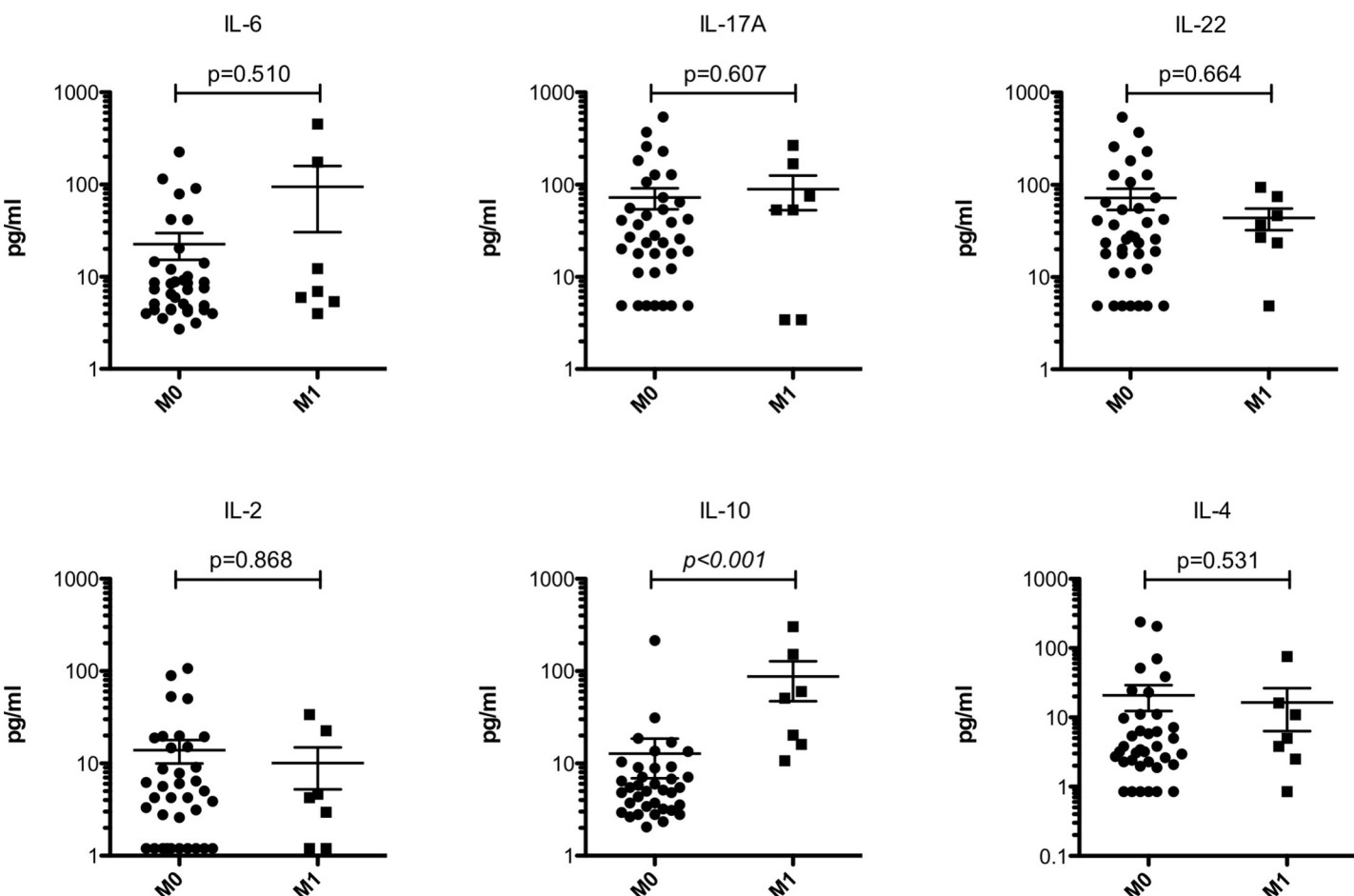

**Fig 4. Systemic IL-10 distinguishes between the presence and absence of distant metastases.** Significant differences (p<0.05) are highlighted in italic. Bars indicate mean with standard error.

## Discussion

This study investigated six pro- and anti-inflammatory cytokines in sera of patients with esophageal adenocarcinomas without neoadjuvant radiochemotherapy, thereby avoiding a possible bias of the immune system due to this therapy.

Levels of IL-6 expression in the tumor environment, where it regulates several signaling pathways that promotes cancer progression, correlates with the prognosis in variety of cancer types [37]. In this study, we demonstrate that increased pro-inflammatory systemic IL-6 levels significantly correlates with the presence of EACs. In line with our data, Hardikar *et al.* reported a twofold increased risk of developing EACs in 411 patients with Barrett's esophagus after dichotomization of systemic IL-6 levels based on median splits [18]. This increment of systemic IL-6 in EAC patients has been reported to be caused by the infiltration of monocytes and macrophages within in the tumor microenvironment as well as by the tumor cells themselves [6,7,38,39]. However, One possibility that can cause increased systemic IL-6 levels may be due to obesity where accumulation of fat leads to a chronic metabolic inflammatory status [40,41]. A risk factor for EACs is obesity [42]. In our cohort, a median BMI of 26.8 kg/m$^2$ was found indicating an obese study population. However, obesity didn't correlate with systemic IL-6 levels and neither did we find any correlation with IL-6 and other clinicopathological

**Table 1. Univariate survival analysis for clinicopathological data.**

| variable | number of patients | dead | survival# | p value* |
|---|---|---|---|---|
| sex | | | | |
| male | 40 (93.0%) | 23 (82.0%) | 18.7 (range 2.3–62.4) | 0.641 |
| female | 3 (7.0%) | 2 (8.0%) | 37.1 (range 19.1–62.1) | |
| T stage | | | | |
| 1 | 9 (20.9%) | 2 (8.0%) | 37.3 (range 3.4–62.4) | 0.051 |
| 3 | 27 (62.8%) | 17 (68.0%) | 15.0 (range 2.3–51.7) | |
| 4 | 7 (16.3%) | 6 (24.0%) | 18.9 (range 3.8–62.1) | |
| N stage | | | | |
| 0 | 11 (25.6%) | 5 (20.0%) | 18.8 (range 2.3–49.5) | 0.221 |
| 1 | 7 (16.3%) | 4 (16.0%) | 31.2 (range 4.0–62.1) | |
| 2 | 11 (25.6%) | 5 (20.0%) | 17.5 (range 4.3–62.4) | |
| 3 | 14 (32.6%) | 11 (44.0%) | 16.8 (range 3.8–48.4) | |
| M stage | | | | |
| 0 | 36 (83.7%) | 19 (76.0%) | 19.1 (range 2.3–62.4) | *0.005* |
| 1 | 7 (16.3%) | 6 (24.0%) | 11.5 (range 3.8–22.9) | |
| UICC | | | | |
| I | 6 (14.0%) | 0 (0.0%) | 18.8 (range 3.4–49.5) | *0.007* |
| II | 7 (16.3%) | 6 (24.0%) | 28.1 (range 2.3–43.1) | |
| III | 23 (53.5%) | 13 (52.0%) | 20.4 (range 4.0–62.4) | |
| IV | 7 (16.3%) | 6 (24.0%) | 11.5 (range 3.8–22.9) | |
| R status | | | | |
| 0 | 30 (69.8%) | 14 (56.0%) | 20.4 (range 3.4–62.4) | *0.005* |
| 1 | 13 (30.2%) | 11 (44.0%) | 13.4 (range 2.3–62.1) | |
| G status | | | | |
| 1 | 3 (7.0%) | 0 (0.0%) | 37.3 (range 18.9–49.5) | 0.146 |
| 2 | 12 (27.9%) | 8 (32.0%) | 12.1 (range 3.4–51.7) | |
| 3 | 28 (65.1%) | 17 (68.0%) | 19.1 (range 2.3–62.4) | |

T stage: describing the size and depth of tumor invasion; N stage: tumor involvement of nearby lymph nodes; M category: distant metastasis; UICC (Union for International Cancer Control): comprising the TNM categories; R: residual tumor at resection margins; G: grading of tumor cells.

* log-rank test

# patients who died within 30days after surgery were excluded. Survival time is presented in months. Significant p values (p<0.05) are highlighted in italic.

factors in our cohort. Blank *et al.* also failed to demonstrate any associations with clinicopathological values and systemic IL-6 levels in a recent analysis [16]. The latter study investigated EACs after neoadjuvant therapy only, hence the systemic immune status might have already been altered. On the contrast, Łukaszewicz-Zając *et al.* found a stronger predictive value of IL-6 as compared to carcinoembryonic antigen (CEA) levels in EAC patients without chemotherapy [17]. This is in line with data obtained from ESCCs and other gastrointestinal malignancies indicating an association with increased tumor stage and poor prognosis [8–13]. In summary, our data support the role of systemic IL-6 levels as potential biomarker in patients with EACs although a distinct stratification for clinicopathological values or an association with overall survival was not found.

Another cytokine involved in the immune host response is IL-2, which is a growth factor for antigen-stimulated T cells and responsible for their clonal expansion. Not only does IL-2 demonstrate a pro-inflammatory effect but it has also been shown that it influences the expansion of anti-inflammatory regulatory T cells and by that stimulates the anti-inflammatory

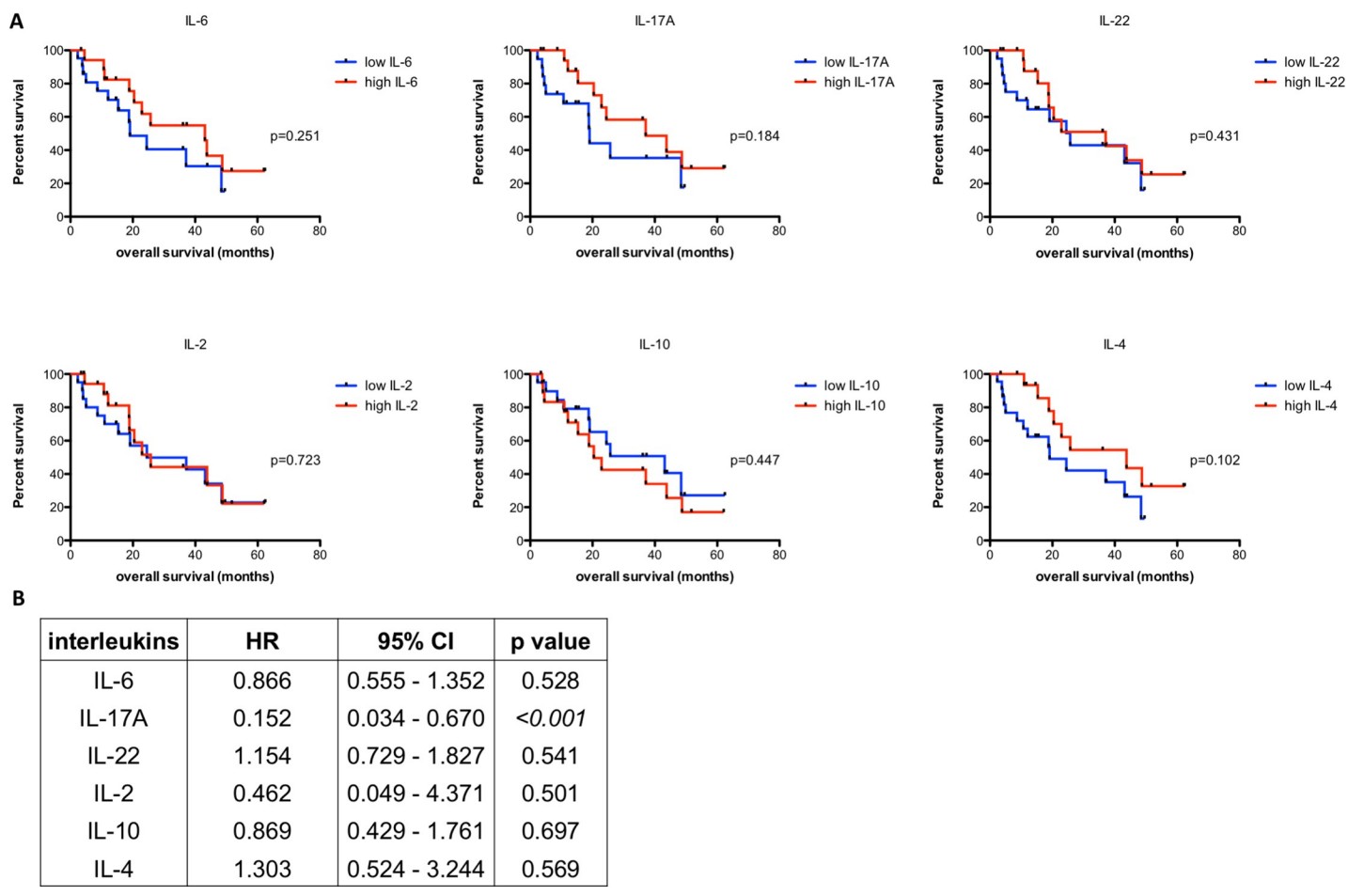

**Fig 5. Influence of systemic interleukins on survival. A)** Univariate survival analysis represented by Kaplan-Meier curves with median as cutoff. **B)** Multivariate cox regression analysis for systemic interleukins. Significant values (p<0.05) are highlighted in italic. Patients who died within 30 days after surgery were excluded from all survival analyses. HR: hazard ratio; CI: confidence interval.

immune response [5]. Interestingly, recombinant IL-2 immunotherapy has been shown to be an effective treatment in malignant melanomas and renal cell carcinomas [43,44]. Though, only few studies have investigated systemic IL-2 in other solid malignancies such as EAC. In our study, neither did we find significant differences between serum levels of healthy controls and EAC patients nor significant associations to clinicopathological features or patient survival. Gabitass *et al*. also did not find a significant difference of systemic IL-2 levels between esophageal cancer patients and controls [21]. In ESCC reduced serum IL-2 levels were found after resection of the tumor as compared to healthy controls but no associations to clinicopathological factors were found [14]. However, in gastric and lung cancer patients increased IL-2 levels were associated with advanced cancer stages [15,45]. Hence, further studies are needed to clarify the role of systemic IL-2 in EACs.

Anti-inflammatory cytokines have been also shown to impact cancer development and patient prognosis. Especially IL-4 and IL-10 seem to be key messengers in immunosuppression. Both cytokines are secreted by activated type 2 T helper cells. Other sources for IL-4 are natural killer cells and eosinophils, while mast cells and regulatory T cells can both secrete IL-10. Among others, reducing the production of CD4+ cells and macrophages causes the anti-inflammatory effect of IL-4. While IL-10 suppresses the pro-inflammatory IL-6 as well as the

production of IL-2 by antigen presenting cells [19,20]. In our cohort, neither did systemic IL-10 nor systemic IL-4 significantly distinguish the presence of EACs from healthy controls. However, slightly, albeit not significantly, increased levels of systemic IL-10 and decreased levels of IL-4 were found in EAC patients as compared to controls. A larger patient collective would have probably led to significant results for both interleukins. This assumption is supported by a study in which increased systemic levels of IL-10 and decreased levels of IL-4 have been found investigating EAC and ESCC together [21]. Conversely, in gastric cancers increased levels of both serum IL-10 and IL-4 have been reported [22–26]. These findings emphasize the use of both systemic IL-10 and IL-4 as potential biomarkers in EACs.

When analyzing systemic IL-10 and IL-4 in regards to clinicopathological data and overall survival, only IL-10 demonstrated to be significantly associated with increased tumor stage and distant metastases. In addition, a rise of systemic IL-10 levels was found going along with an increase of UICC stage. The presence of high systemic IL-10 levels is similar to the report by de Vita et al., where higher levels of IL-10 were observed in patients with advanced or metastatic gastrointestinal malignancies as compared to healthy controls [22]. In addition, a meta-analysis by Zhao et al. identified serum IL-10 as a negative prognostic marker in colon and gastric cancers as well as in other gastrointestinal malignancies although this study didn't include esophageal cancers [27]. Interestingly, esophageal cancer patients with late stage tumors harbor a greater density of regulatory T cells (CD4[+]CD25[high]) in their peripheral blood as opposed to early stage cancers. High proportions of these systemic regulatory T cells were associated with lower survival rates [46]. In a recent study, we reported an increased number of tissue resident IL-10 producing FOXP3[+] regulatory T cells in EAC patients, which were significantly associated with reduced survival rates. Moreover, the correlation on patients' survival was not only found for IL-10 within the tumor but also for the unaltered mucosa close to the resection margin representing a local immunological field effect [28]. Together with the current results, one might hypothesize, that this anti-inflammatory environment extends to the blood stream and helps malignant cells to evade host response and influences the oncogenetic and metastatic ability of neoplasms. Of note, IL-10 can have pleiotropic functions, which might depend on the cellular source. Nevertheless, we suggest that manipulating regulatory T cells to produce less IL-10 might be a potential treatment option, which has already been tested in other malignancies with promising results [47,48]. However, a correlation with overall survival was neither found for systemic IL-10 nor systemic IL-4 in this study. To conclude, IL-10 demonstrated to be strongly associated with advanced stages of EACs but further investigations are needed to evaluate its influence on patient survival.

Other cytokines, which are usually present in chronic inflammation, are IL-22 and IL-17. IL-22 is produced by innate lymphoid cells, IL-17[+] and IL-22[+] cells and is involved in both wound healing and tumor development [29–31]. Inhibitory as well as tumor promoting effects have been reported [32]. IL-17[+] cells are another subset of T helper cells, which primarily secrete IL-17A, IL-17F but also IL-22. They are thought to be involved in immunoregulation and have been found in the development of inflammation-related tumors. IL-17[+] cells seem to both bare anti- and pro-inflammatory properties [33,34]. An increase of systemic IL-22[+] and IL-17[+] cells in the progression of gastric carcinoma as well as in colorectal cancer has been reported [49,50]. Interestingly, systemic IL-17A was significantly decreased in EAC patients as compared to controls while systemic IL-22 didn't significantly differ between both groups. However, only increased systemic IL-17A levels were associated with shortened survival. No other significant alterations in relation to clinicopathological factors were found for IL-17A or IL-22. The prognostic significance of systemic IL-17 was also reported in a previous study demonstrating increased proportion of Th17[+] cells within the peripheral blood and tumor tissues of esophageal cancer patients. In addition, the amount of Th17[+] cells increased from early

to advanced tumor stages [51]. A limit of this study is that the systemic IL-17 levels were not directly analyzed and the authors instead showed the Th17 cells differentiation-related cytokines IL-23, IL-1β, and IL-6 in the local tumor tissue. Whereas, other studies revealed that IL-17A can enhance the tumor killing capability via humoral cell-mediated induction of immunogenic antibody and cytolytic molecules in ESCC [52]. Though, the role of circulating IL-17 or IL-22 in esophageal cancers is not yet defined [36] and further studies are needed to clarify the role of IL-17A and IL-22 in EACs.

A general limitation in analyzing systemic interleukins lies in the high number of confounding factors potentially influencing systemic interleukin levels. For example, Cook *et al.* recently described a potential link between excessive obesity and cigarette smoking with systemic inflammation [36]. Also, other environmental factors like alcohol or the composition of the microbiota might alter systemic inflammation. These limitations have to be kept in mind when interpreting our data. Another limitation is the lack of early stage esophageal carcinomas in the study. However, most of the patients with EACs are diagnosed at an already advanced stage due to the absence of symptoms. This emphasizes the urgent need for valid systemic biomarkers to detect patients in early stages and by that improve overall survival.

In summary, increased systemic IL-6 and lower IL-17A levels distinguish EAC patients from healthy controls. Importantly, systemic IL-10 levels correlate with the increase of the local tumor burden (T stage) and are also significantly in patients with distant metastasis. However, only IL-17A levels influenced survival in multivariate analysis. Hence, circulating IL-10 might help to identify EAC patients with advanced disease and systemic IL-17A might indicate a limited prognosis.

## Supporting information

**S1 Fig. Systemic IL-10 levels rise with increasing UICC stages.** For better visualization p values are only given for significant tests (p<0.05). Bars indicate mean with standard error. (TIFF)

**S1 Table. Spearman correlation matrix.** T stage: describing the size and depth of tumor invasion; N stage: tumor involvement of nearby lymph nodes; M category: distant metastasis; UICC (Union for International Cancer Control): comprising the TNM categories; R: residual tumor at resection margins; G: grading of tumor cells. Significant p values (p<0.05) are highlighted in italic. Correlation coefficient is presented in parenthesizes. (DOCX)

## Author Contributions

**Conceptualization:** Karl-Frederick Karstens, Samuel Huber, Penelope Pelczar.

**Data curation:** Penelope Pelczar.

**Formal analysis:** Karl-Frederick Karstens, Jan Kempski, Anastasios D. Giannou, Erik Freiwald, Nicola Gagliani.

**Investigation:** Karl-Frederick Karstens, Jan Kempski, Anastasios D. Giannou, Penelope Pelczar.

**Resources:** Jakob R. Izbicki, Ansgar W. Lohse, Samuel Huber.

**Supervision:** Nicola Gagliani, Samuel Huber.

**Writing – original draft:** Karl-Frederick Karstens, Jan Kempski, Anastasios D. Giannou.

**Writing – review & editing:** Erik Freiwald, Matthias Reeh, Michael Tachezy, Jakob R. Izbicki, Ansgar W. Lohse, Samuel Huber, Penelope Pelczar.

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
