## [Decision Letter · Decision Letter 0]

2 Mar 2020

PONE-D-20-00015

Systemic interleukin 10 and 17A levels indicate advanced stages and reduced survival in esophageal adenocarcinomas

PLOS ONE

Dear Dr. Huber,

Thank you for submitting your manuscript to PLOS ONE. After careful consideration, we feel that it has merit but does not fully meet PLOS ONE’s publication criteria as it currently stands. Therefore, we invite you to submit a revised version of the manuscript that addresses the points raised during the review process.

ACADEMIC EDITOR: 

1. The authors addressed that " Blood samples were collected  from  the  biobank  of  the  Department  of  Surgery." (lines 109-110) The timing of blood drawing (at diagnosis, before treatment or during treatment, after treatment) in the EAC group has not been mentioned. 

2. The company, location, country information of experiment kits, and machines/devices are not clearly described. 

3. Multivariate analysis should include clinicopathological features to adjust their independent role. 

4. Lines 310-312, " However, a trend  of increased levels of systemic IL-10 and decreased levels of IL-4 were found in EAC patients as compared to controls." But the p-value of IL-10 show near no significance. This sentence should be revised. 

We would appreciate receiving your revised manuscript by Apr 16 2020 11:59PM. To enhance the reproducibility of your results, we recommend that if applicable you deposit your laboratory protocols in protocols.io, where a protocol can be assigned its own identifier (DOI) such that it can be cited independently in the future. For instructions see: http://journals.plos.org/plosone/s/submission-guidelines#loc-laboratory-protocols

We look forward to receiving your revised manuscript.

Kind regards,

Jason Chia-Hsun Hsieh, M.D. Ph.D

Academic Editor

PLOS ONE

Additional Editor Comments (if provided):

1. The authors addressed that " Blood samples were collected from the biobank of the Department of Surgery." (lines 109-110) The timing of blood drawing (at diagnosis, before treatment or during treatment, or after treatment) in the EAC group has not been mentioned.

2. The company, location, country information of experiment kits, and machines/devices are not clearly described.

3. Multivariate analysis should include clinicopathological features to adjust their independent role.

4. Lines 310-312, " However, a trend of increased levels of systemic IL-10 and decreased levels of IL-4 were found in EAC patients as compared to controls." But the p-value of IL-10 show near no significance. This sentence should be revised.

Journal Requirements:

2) PLOS requires an ORCID iD for the corresponding author in Editorial Manager on papers submitted after December 6th, 2016. Please ensure that you have an ORCID iD and that it is validated in Editorial Manager. To do this, go to ‘Update my Information’ (in the upper left-hand corner of the main menu), and click on the Fetch/Validate link next to the ORCID field. This will take you to the ORCID site and allow you to create a new iD or authenticate a pre-existing iD in Editorial Manager. Please see the following video for instructions on linking an ORCID iD to your Editorial Manager account: https://www.youtube.com/watch?v=_xcclfuvtxQ

Reviewers' comments:

Reviewer's Responses to Questions

**Comments to the Author**

1. Is the manuscript technically sound, and do the data support the conclusions?

Reviewer #1: Yes

Reviewer #2: Partly

Reviewer #3: Partly

2. Has the statistical analysis been performed appropriately and rigorously? 

Reviewer #1: Yes

Reviewer #2: I Don't Know

Reviewer #3: Yes

3. Have the authors made all data underlying the findings in their manuscript fully available?

Reviewer #1: Yes

Reviewer #2: Yes

Reviewer #3: No

4. Is the manuscript presented in an intelligible fashion and written in standard English?

Reviewer #1: Yes

Reviewer #2: Yes

Reviewer #3: Yes

5. Review Comments to the Author

Reviewer #1: 1. The manuscript makes a contribution to the knowledge of systemic inflammatory interleukins in patients with esophageal adenocarcinoma.

2. Page 2 Line 42, the sentence “we wanted to investigated” should be “we wanted to investigate”.

3. May suggest English editing if necessary.

Reviewer #2: I am not familiar with this area about relationship between EAC and IL. But I read this manuscript with interesut. Please elaborate on why circulating IL-10 and IL-17A can help identify patients with advanced disease and may serve as potential targets for future tumor treatment.

Reviewer #3: This is an interesting proposal “Systemic interleukin 10 and 17A levels indicate advanced stages and reduced survival in esophageal adenocarcinomas” designed to evaluate the presence of systemic inflammatory interleukins (IL) and their impact on the prognosis. Of esophageal adenocarcinomas patient. The investigators found the significant higher systemic IL-6 and lower IL-17A in EAC patients and IL-10 levels has trend increase in EAC patients. There is a correlation between increases IL-10 with an increased T stage. Also, IL-10 levels were higher in patients with distant metastasis.

The investigators are doing measurement of Interleukins study in EAC patients with different conditions and need to provide clear data analysis to support their conclusion, such as survival issue. Also, to figure out the possible mechanism of IL10 as marker in EAC tumor, authors should provide the detail of the localization of IL10/17 and foxp3/CD25 in EAC with T stage or with distance metastasis. Other issues are listed as following:

Specific comments.

Major points

• Title: The title seems not appropriate. IL-10 level correlates TMN stage but not survival. IL-17 showed correlation with survival but seems not convincing enough.

• Line 57: It is not shown the correlation between IL-17 and advanced disease

• 71: Rationales of selecting these 6 markers is unclear. Why other cytokines such as IL-8, IFNg are not selected?

• 154: There is no trend in IL-10 level because p value is over 0.1

• 311: same as line 154, there is no trend of increase in IL-10

• 317: same as line 154, there is no trend of increase in IL-10

• 334: This might be saying too much. Data in this study shows a potential of IL-10 in diagnosis but it doesn't mean antagonizing IL-10 will have a therapeutic potential. IL-10 has been known to have pleiotropic function: anti-inflammatory effect and immunostimulatory effect. In the field of cancer therapy, pegylated IL-10 is used to induce cancer immunity that is opposite to antagonizing IL-10. Then it might be true that increase of IL-10 indicates increased inflammation caused by cancer but not necessarily means inhibition of IL-10 has a therapeutic effect.

• 369: I am not sure what the author is trying to say as a limitation.

• 376: same as line 154, there is no trend of increase in IL-10 in EAC patients

• 380: IL-17 is not correlates with tumor stage so it could not be said

• 381: The data of this paper showed potential in diagnosis or prognosis rather than therapeutics

Minor point

• According to Method, the sample used for assay is not serum but plasma but “serum” is used in the body.

6. PLOS authors have the option to publish the peer review history of their article (what does this mean?). If published, this will include your full peer review and any attached files.

Reviewer #1: No

Reviewer #2: No

Reviewer #3: No

---

## [Author Response · Author response to Decision Letter 0]

13 Mar 2020

Please find attached the "Reply to reviewers".

---

## [Editor Report · Decision Letter 1]

2 Apr 2020

Systemic interleukin 10 levels indicate advanced stages while interleukin 17A levels correlate with reduced survival in esophageal adenocarcinomas

PONE-D-20-00015R1

Dear Dr. Huber,

We are pleased to inform you that your manuscript has been judged scientifically suitable for publication and will be formally accepted for publication once it complies with all outstanding technical requirements.

With kind regards,

Jason Chia-Hsun Hsieh, M.D. Ph.D

Academic Editor

PLOS ONE

Additional Editor Comments (optional):

The authors responded to all of the questions adequately.
---

## [Editor Report · Acceptance letter]

6 Apr 2020

PONE-D-20-00015R1 

Systemic interleukin 10 levels indicate advanced stages while interleukin 17A levels correlate with reduced survival in esophageal adenocarcinomas 

Dear Dr. Huber:

I am pleased to inform you that your manuscript has been deemed suitable for publication in PLOS ONE. Congratulations! Your manuscript is now with our production department. 

With kind regards,

on behalf of

Dr. Jason Chia-Hsun Hsieh 

Academic Editor

PLOS ONE